# *Polygonatum sibiricum* Saponin Prevents Immune Dysfunction and Strengthens Intestinal Mucosal Barrier Function in Cyclophosphamide-Induced Immunosuppressed BALB/c Mice

**DOI:** 10.3390/foods13060934

**Published:** 2024-03-19

**Authors:** Dongyun Zhao, Huanhuan Liu, Chunhong Yan, Yue Teng, Yue Zou, Xiaomeng Ren, Xiaodong Xia

**Affiliations:** State Key Laboratory of Marine Food Processing and Safety Control, National Engineering Research Center of Seafood, School of Food Science and Technology, Dalian Polytechnic University, Dalian 116034, China; 211710832000922@xy.dlpu.edu.cn (D.Z.); 20173083200048@xy.dlpu.edu.cn (H.L.); chyan@dlpu.edu.cn (C.Y.); 20173083200027@xy.dlpu.edu.cn (Y.T.); 1717010103@xy.dlpu.edu.cn (Y.Z.); renxm@dlpu.edu.com (X.R.)

**Keywords:** *Polygonatum sibiricum*, saponin, immunomodulation, intestinal mucosal barrier, cyclophosphamide

## Abstract

The aim of this study was to explore the immunomodulatory effect of *Polygonatum sibiricum* saponin (PS) in a cyclophosphamide-induced (Cy) immunosuppression mice model. Oral administration of PS by gavage effectively alleviated weight loss caused by Cy and increased the index of immune organs. PS promoted the proliferation of splenic lymphocytes and T cell subsets (CD3^+^, CD355^+^, CD4^+^/CD8^+^) and relieved the xylene-induced inflammatory response and Cy-induced increase of serum hemolysin. Moreover, PS increased serum levels of lactate dehydrogenase and acid phosphatase. PS elevated serum level of cytokines and immunoglobulins (TNF-α, IFN-γ, IL-4, IL-6, IL-β, SIgA, and IgG) and the expression of mRNA of *IL-10*, *TNF-α*, and *IL-6* in the spleen. Increased mRNA expression of tight junction protein (*ZO-1*, *Mucin2*, *Occludin*) expression and protein expression of IL-6/MyD88/TLR4 in the small intestine showed that PS exhibited a restorative effect on intestinal mucosal injury caused by cyclophosphamide. Oral PS prevented Cy-induced decline in leukocytes, red blood cells, lymphocytes, hemoglobin concentrations, and neutrophils, providing evidence for alleviating hematopoietic disorders. In addition, PS increased SOD and NO levels, reduced MDA levels, and improved oxidative damage in the liver. These findings demonstrate that PS has the potential to be developed as a supplemental agent for alleviating immunosuppression caused by chemotherapeutic agents.

## 1. Introduction

According to the WHO, cancer has become the number one killer of humans in recent years [1]. Cyclophosphamide (Cy) is widely used in cancer treatment and is effective in restricting cancer cell growth. It can also have adverse effects on the body. Cy is inactive in vitro and functions in vivo primarily through the hydrolysis by hepatic P450 enzymes into aldehyde phosphamide [2], which interferes with DNA and RNA function. Cy could affect the cell cycle, thereby inhibiting the proliferation of T and B lymphocytes [3], resulting in immune suppression [4,5]. At present, many natural bioactive substances are currently explored to treat immune function decline [6,7,8]. It has been shown that Cordyceps sinensis spore polysaccharides improve cyclophosphamide-induced immunosuppression in mice by promoting splenocyte proliferation and enhancing macrophage activity [9]. The combination of saffronin and sorafenib in the treatment of liver cancer activates apoptosis and restores liver function [5]. G-Rh2 can have synergistic anti-tumor effects with Cy by regulating fatty acid metabolism in NSCLC mice [10]. In addition, there are data to suggest that wheat bran polyphenols ameliorate DSS-induced ulcerative colitis in mice by inhibiting the MAPK/NF-κB inflammasome pathway [11], *Polygonatum sibiricum* saponin has been shown to reduce blood glucose by regulating the intestinal flora [12], and dandelion exhibits a protective effect against liver fibrosis and oxidative stress in rats [13]. These results suggest that the search for functional active substances in natural products has attracted widespread attention from nutritionists and biomedical researchers [7,14]. Saponins, as a type of natural active substance, have been shown to improve physiological health [15], including the sedative effect of jujube saponins on sleep aids [15] and the immunomodulatory and antitumor effects of astragalus saponins [10]. These immune-enhancing effects could be evidenced by the changes in cytokines and immunoglobulins and increased T lymphocyte proliferation [16] and differentiation, which could be achieved through modulating pathways such as the p62/Keap1/Nrf2 pathway [17]. Currently, a variety of food and drug homologous substances have been demonstrated to exert immunomodulatory effects, such as ginseng [18], red dates, Chinese wolfberry, yams [19], and hawthorn [20].

*Polygonatum sibiricum* is a species of lily in the family Lilyaceae, found mainly in the northern temperate zone. *Polygonatum sibiricum* has been listed in the Medicine and Food Homologous Catalog in China. It contains a variety of active ingredients: polysaccharides [21], flavonoids [21], saponins [22], lignans, and other trace elements [23]. At present, it has been known that *Polygonatum sibiricum* has a variety of beneficial functions, including anti-depression [24], anti-diabetes [12], anti-obesity [25], anti-osteoporosis [26], anti-atherosclerosis [27], and anti-cell proliferation [28]. Saponin is one of the main active substances in *Polygonatum sibiricum,* and it has a variety of biological activities, including immunomodulatory [29], anticancer [30], anti-colitis [31], and anti-diabetes functions [12]. At present, whether PS could reduce the immunosuppression and intestinal damage caused by Cy remains unexplored. Therefore, this study was designed to examine the effect of *Polygonatum sibiricum* saponin on immune function and the intestinal barrier in a Cy-induce immunosuppression mice model.

## 2. Material and Methods

### 2.1. Reagents

Cyclophosphamide was purchased from Baxter Medical Supplies Trading Co., Ltd. (Shanghai, China). Levamisole hydrochloride was purchased from Maclin Biochemical Technology Co., Ltd. (Shanghai, China). Neutral red was purchased from Regan Biotechnology Co., Ltd. (Beijing, China). CCK-8, Hank’s, NO, and JC-1 kits were obtained from Beyotime Biotechnology Co., Ltd. (Shanghai, China). RPMI1640 was bought from Gibco (BRL Co., Ltd., Gaithersburg, MD, USA). Red blood cell lysate and trypan blue were purchased from Beijing Solarbio Biotechnology Co., Ltd. (Beijing, China). Concanavalin A was purchased from Yuanye Biotechnology Co., Ltd. (Shanghai, China). Lipopolysaccharides were bought from Sigma (St. Louis, MO, USA). Malondialdehyde (MDA), superoxide dismutase (SOD), nitric oxide (NO), acid phosphatase (ACP), lactate dehydrogenase (LDH), and alanine aminotransferase (ALT) kits were purchased from Nanjing Institute of Bioengineering (Nanjing, China). IL-4 (interleukin-4), IL-6 (interleukin-6), TNF-α (tumor necrosis factor-α), INF-γ (interferon-γ), and IgG (immunoglobulin G) were purchased from Shanghai Enzyme-linked Biotechnology Co., Ltd. (Shanghai, China). SIgA (secretory immunoglobulin A) was purchased from Jonln Biotechnology Co., Ltd. (Shanghai, China). Reverse transcription kits and qPCR kits were purchased from AG Biotechnology Co., Ltd. (Shanghai, China).

### 2.2. Preparation of Polygonatum sibiricum Saponin

Crude *Polygonatum sibiricum* saponin was extracted and purified according to the methods described previously [32]. Powder of *Polygonatum sibiricum* (50 g) was extracted with 1, 250 mL of 80% ethanol (containing 3% cellulase and pectinase) with ultrasound for 1 h. The extraction was repeated three times and the pooled extract was filtered and concentrated under reduced pressure to remove the ethanol and water. The residue was extracted with saturated n-butanol and purified with D101 macroporous adsorbent resin column chromatography by eluting with water and 70–90% ethanol in turn. The entire ethanol fraction was gathered and concentrated to obtain *Polygonatum sibiricum* saponins (PS). The extraction rate of saponins from *Polygonatum sibiricum* was 27%.

### 2.3. Animals and Experiments Design

A total of ninety male Balb/c mice (4 weeks old; 20 ± 2 g) were purchased from Liaoning Changsheng Biotechnology Co., Ltd. (Benxi, China). The Animal Ethics Committee of Dalian University of Technology granted approval for all animal studies, which were conducted in compliance with the Guidelines of the National Institute for Animal Studies (Animal Experiment License: DLPU2022057). Each rectangular cage housed five mice and was kept under conventional laboratory circumstances (temperature 23 ± 2 °C, relative humidity 55 ± 5%, and 12 h light/dark cycle). The mice were fed a rodent maintenance feed from Liaoning Changsheng Biotechnology Co., Ltd. Following a one-week acclimation period, the mice were subsequently allocated into six groups: the control group, the Cy group, the low-dose PS intervention with Cy group (LPS + Cy), the middle-dose PS intervention with Cy group (MPS + Cy), the high-dose PS intervention with Cy group (HPS + Cy), and the levamisole hydrochloride intervention with Cy group (Positive + Cy) (*n* = 15). Mice in the control and Cy groups were intragastrically gavaged with sterile distilled water, while mice in LPS + Cy, MPS + Cy, HPS + Cy, and Positive + Cy groups were gavaged with 30 mg/kg/d PS, 50 mg/kg/d PS, 70 mg/kg/d PS, and 100 mg/kg/d levamisole hydrochloride, respectively, during the first 10 days. From day 11, the mice in the Cy, LPS + Cy, MPS + Cy, HPS + Cy, and Positive + Cy groups were intraperitoneally injected with 80 mg/kg cyclophosphamide for five days. The mice were weighed daily during the trial, and food intakes were recorded every day. At the end of the intervention, mice were fasted overnight and provided with sterile water. Mice were intraperitoneally injected with sodium pentobarbital (80 mg/kg) for anesthesia. Blood samples were taken by retrobulbar puncture and then mice were euthanized by cervical dislocation. Liver, spleen, thymus, small intestine, and small intestine contents were then collected for further analysis.

### 2.4. Organ Index

After mice were euthanized, the thymus, spleen, and liver of each mouse were aseptically removed and weighed, and the following index was calculated. The organ indexes were computed using the subsequent formula:(1)$$\mathrm{Thymus}\;\mathrm{index}\;(\%)=\frac{thymus\;weight(g)}{body\;weight(g)}\;\times 100\%$$
(2)$$\mathrm{Spleen}\;\mathrm{index}\;(\%)=\frac{Spleen\;weight(g)}{body\;weight(g)}\;\times 100\%$$
(3)$$\mathrm{Liver}\;\mathrm{index}\;(\%)=\frac{Liver\;weight(g)}{body\;weight(g)}\;\times 100\%$$

### 2.5. Xylene-Induced Inflammatory Response

To examine the acute inflammatory reactions caused by xylene in mice, 20 μL of xylene was uniformly applied to the surface of the left ear on the 15th day. The vernier caliper was used to measure the thickness of both the left and right ear in order to compare the degree of swelling on the two ears.

### 2.6. Serum Hemolysis Value (HC_50_) Assay

On the third day of cyclophosphamide injection, 100 μL of 5% chicken red blood cells (CRBC) was intraperitoneally injected into mice for two days. Blood samples were obtained from the orbital venous plexus after the final injection of 5% chicken erythrocytes and centrifuged at 5000 rpm/min at 4 °C for 5 min. The serum was diluted 100 times with normal saline. Equal amounts (0.5 mL) of the diluted serum, 5% CRBC, and 10% guinea pig serum were mixed at 37 °C for 30 min. The mixture was immediately placed on ice for 15 min to halt the reaction. The suspension of the blank serum was prepared by adding 0.25 mL of 5% chicken erythrocytes and 0.5 mL of 10% guinea pig serum to 1.25 mL of normal saline. The resulting mixture was centrifuged and the absorbance was measured in a microplate reader at 540 nm [33].

### 2.7. Hematological Analysis

Blood samples were obtained and treated with EDTA to prevent coagulation. A hematology analyzer (BC-5000Vet model from Mindray, Shenzhen, China) was used to measure blood counts. The analysis included white blood cells (WBC), red blood cells (RBC), hemoglobin (HGB), hematocrit (HCT), platelet count (PLT), lymphocytes (Lym#), and granulocytes (Gran).

### 2.8. Histological Analysis

The spleen and small intestinal tissues were subjected to fixation using a 4% neutral paraformaldehyde solution for a duration of 24 h. Following fixation, the tissues were processed for paraffin embedding, sectioned, and subsequently stained with hematoxylin and eosin (H&E). The thickness of the mucus and the presence of goblet cells in the intestine were assessed using periodic Alcian blue staining (AB/PAS). Microscopic examination was conducted to observe the histopathological features using the Nikon Eclipse TI-S microscope, Tokyo, Japan.

### 2.9. Measurement of Immune-Related Cytokines in the Serum

The blood from mice was obtained and centrifuged at 3500 rpm for 10 min at 4 °C. The supernatant was separated and stored at −80 °C for measurement. The levels of IL-6, IL-1β, IL-4, TNF-α, IFN-γ, SIgA, and IgG were measured by enzyme-linked immunoassay kit (ELISA) methods. Acid phosphatase (ACP) and Lactate Dehydrogenase (LDH) levels were determined according to the kit instructions.

### 2.10. Determination of Antioxidant Levels in Liver Tissue

The liver tissues were precisely weighed and blended in 0.9% normal saline at a ratio of 1:9 (weight/volume). The homogenate was centrifuged and the supernatant was collected. The protein concentration of liver tissue was determined using the BCA kit and the levels of alanine aminotransferase (ALT), malondialdehyde (MDA), superoxide dismutase (SOD), and nitric oxide (NO) were measured in accordance with the manufacturer’s instructions.

### 2.11. Splenocyte Proliferation Assay

The splenic lymphocyte proliferation experiment was performed according to a previously described method [34]. The spleen tissues were aseptically collected and the connective tissue surrounding the spleen was excised. Subsequently, 3 mL of Hank’s solution was added to the tissues and they were homogenized. The cell suspension underwent sorting using a 70 μm sieve (BD Biosciences), and red cell lysis buffer was added to lyse red blood cells for 15 min. RPMI1640 was added after complete lysis and the solution was centrifuged to obtain cells for subsequent experiments.

### 2.12. Lymphocyte Proliferation

The splenic lymphocyte proliferation assay was determined as previously described [35]. The CCK-8 assay was used to assess the proliferation of T and B splenic lymphocytes induced by ConA and lipopolysaccharides. The cell concentration per well was 1 × 10^6^ cells/mL. An equal volume of Con A (20 μg/mL) or lipopolysaccharides (10 μg/mL) was added to the well of 96-well plates. RPMI1640 cultured splenocytes served as a control group, and RPMI1640 serves as a blank control. The splenocytes were cultured for 48 h at 37 °C in 5% CO_2_. After 48 h, 10 μL of CCK-8 solution was added into each well and the solutions were subsequently incubated for an additional 4 h, followed by measurement of the absorbance value at 450 nm.

### 2.13. Flow Cytometric Analysis

The splenic lymphocytes were adjusted to 1 × 10^6^ cells/mL. One microliter each of APC-CD3^+^, PE-CD4^+^, FITC-CD8^+,^ and PE-CD355^+^ (Biolegend, USA) was added into 100 μL of cell suspension, which was then incubated for 1 h on ice in the dark after gentle mixing. Finally, the detection of lymphocytes was performed by flow cytometry (Cyto Flex, Beckman Coulter, Inc., Brea, CA, USA).

### 2.14. Western Bolt Analysis

Small intestine tissues were homogenized in RIPA buffer with 1% PMSF and 2% Protease phosphatase inhibitors (Beyotime, Shanghai, China), centrifuged at 12,000× *g* at 4 °C for 10 min, and the protein concentration was determined by BCA assay kits (Solarbio, Beijing, China). The total protein was separated by SDS-PAGE and transferred to polyvinylidene fluoride (PVDF) membranes. The membranes were subjected to blocking using a 5% (*w*/*v*) solution of skim milk powder for a duration of 90 min at room temperature. Subsequently, the membranes were incubated overnight at 4 °C with the primary antibody. The membrane was washed three times with TBST for 15 min each and incubated with secondary antibodies for 1 h. After washing three times, the membranes were treated with a chemiluminescent substrate and bands were developed using an ECL Chemiluminescence Detection Kit and were analyzed with the Bio-Rad Image Analysis System (Berkeley, CA, USA).

### 2.15. Quantitative RT-PCR Analysis

The extraction of total RNA from spleen samples was performed using trizol reagent (Shanghai Sangon Biotechnology Co., Ltd., Shanghai, China) and cDNA samples were produced using the PrimeScript™ RT Kit (cDNA Eraser). The mRNA expression levels were evaluated using the quantitative RT-PCR method with primers listed in Table 1, specifically employing the TB Green^®^ Premix Ex Taq™ II Kit. The gene expression levels of the target genes were normalized to the reference levels of GAPDH expression and computed using the 2^−∆∆Ct^ method.

### 2.16. Statistical Analysis

GraphPad Prism 9.00 software was used for Statistical analysis (La Jolla, CA, USA). The data were presented in the form of mean values accompanied by the standard error of the mean (SEM). The statistical significance of the differences was established by the utilization of analysis of variance and Tukey’s multiple comparison test, with a significance level of *p* < 0.05.

## 3. Results

### 3.1. Body Weight and Organ Index Change of Mice

Before Cy treatment, the weight of mice in all groups continued to increase, indicating that PS had no toxic side effects on mice (Figure 1A). After Cy treatment, except for the control group, the weight of mice in all groups decreased, while mice in Cy + LPS, Cy + MPS and Cy + HPS groups showed les weight loss compared to the Cy group. The average food intake of mice in each group is shown in the Figure 1B. The control group kept a steady food intake, while the food intake in the Cy group was in a decline. Mice in the Cy + LPS, Cy + MPS, and Cy + HPS groups all ate more than the Cy group. The spleen, liver, and thymus were organs directly related to the immune response, and we assessed changes in organ indexes. As shown in Figure 1C–E, compared to the control group, the spleen index, liver index, and thymus index of mice in the Cy group all decreased, while indices in Cy + LPS, Cy + MPS, and Cy + HPS groups had increased to varying degrees.

### 3.2. PS Supplementation Improved Histopathological Alterations in Cyclophosphamide-Induced Immunosuppressed Mice

HE sections of mice spleen in each group were observed under a light microscope. As shown in Figure 2, the structure of the spleen of mice in the control group was normal, with lymphocytes densely arranged and showing intact structure of red pith and white pulp. The lymphocytes in the model group were sparse, the pulp area became larger. After PS intervention, the marginal areas of red pulp and white pulp of mouse spleen tissue were restored and lymphocytes increased, indicating that PS had a recovery effect on spleen damage caused by cyclophosphamide.

### 3.3. PS Modulated Hemolysin Levels in Cyclophosphamide-Induced Immunosuppressed Mice

Hemolysin is a protein that is produced in response to the stimulation of a foreign antigen, and the measurement of serum hemolysin levels can reflect the humoral immune status of the body [36]. As shown in Figure 3A, compared to the control group, the *HC*_50_ in the Cy group was significantly reduced (*p* < 0.05), indicating that cyclophosphamide induced immunosuppression. The *HC*_50_ in Cy + LPS, Cy + MPS, and Cy + HPS groups were significantly improved. The results showed that PS can enhance humoral immunity by raising serum hemolysin values.

### 3.4. PS Regulated Lactate Dehydrogenase and Acid Phosphatase Levels in Cyclophosphamide-Induced Immunosuppressed Mice

Acid phosphatase (ACP) and lactate dehydrogenase (LDH) are signature enzymes of macrophages that can reflect macrophage activity in mice. As shown in Figure 3B,C, after cyclophosphamide treatment, the serum of ACP and LDH in mice were significantly reduced (*p* < 0.01). Compared with the Cy group, the activity of ACP and LDH was significantly increased in most of the PS treatment group (*p* < 0.05). LDH activity showed an increased trend in the HPS group with no significant difference compared to the Cy treated group.

### 3.5. PS Alleviated Xylene-Induced Inflammation in Cyclophosphamide-Induced Immunosuppressed Mice

Xylene-induced ear swelling in mice was used to simulate the pathological state of acute inflammation in the body. As shown in Figure 3D, the ear swelling rate in the control group was 8.62%, while the Cy group reached 83.57% (*p* < 0.0001). The Cy + LPS, Cy + MPS, and Cy + HPS groups significantly reduced the degree of ear swelling caused by xylene. The ear swelling rates in the LPS, MPS, and HPS groups were 35.97%, 11.46%, and 34.82%, respectively, which indicated that oral PS reduced xylene-induced inflammatory responses.

### 3.6. Recovery Effect of PS on Blood Counts in Cyclophosphamide-Induced Immunosuppressed Mice

The protective effect of PS on hematopoietic damage caused by cyclophosphamide was evaluated by detecting the number of WBC, RBC, HGB, HCT, PLT, LYM, and GRAN in whole blood. As shown in Table 2, compared to the control group, the number of WBC, HGB, HCT, PLT, and LYM in the Cy group was sharply decreased (*p* < 0.0001), and the number of RBC and Gran in the Cy group also decreased (*p* < 0.05). The number of WBC, HGB, HCT, PLT, and Lym in Cy + LPS, Cy + MPS, and Cy + HPS groups was significantly increased (*p* < 0.05) compared to the Cy group, and the number of RBC and Gran in the three treatment groups increased significantly. These demonstrated that PS reduced the decline in hematopoietic function caused by cyclophosphamide.

### 3.7. PS Supplementation Improved Splenic Lymphocyte Proliferation in Cyclophosphamide-Induced Immunosuppressed Mice

To determine the effect of PS on immune cells in Cyclophosphamide-induced immunosuppressed mice, splenic lymphocyte proliferation and lymphocyte subsets were analyzed. As shown in Figure 4A,B, the proliferation of B lymphocytes caused by lipopolysaccharides and the proliferation of T lymphocytes induced by ConA were measured. In comparison to the control group, the proportion of lymphocyte proliferation decreased in the cyclophosphamide treatment group (*p* < 0.0001) and PS at three different concentrations all improved lymphocyte proliferation in immunosuppressed mice to varying degrees (*p* < 0.05 or *p* < 0.01). As shown in Figure 4C–I, cyclophosphamide decreased the ratio of CD3^+^ and CD355^+^ and the CD4^+^/CD8^+^ ratio decreased. After treatment with PS, the number of CD3^+^ T cells and CD355^+^ T cells increased, the ratio of CD3^+^ T cells in LPS, MPS, and HPS increased by 3.8%, 6.4%, and 2.9%, respectively, and the ratio of CD355^+^T cells in the three groups increased by 7.1%, 18%, and 6.8%, respectively. At the same time, the CD4^+^/CD8^+^ ratio in the treatment group was also improved compared to the Cy group.

### 3.8. PS Modulated Inflammatory Factors and Immunoglobulin in Cyclophosphamide-Induced Immunosuppressed Mice

The immunomodulatory effect of PS was assessed by measuring cytokine and immunoglobulin levels in the serum. As shown in Figure 5A–E, compared to the control group, the levels of IFN-γ, TNF-α, IL-4, IL-6, and IL-1β were all reduced in the Cy group, while their levels increased in the PS-treated groups, which showed that PS could enhance the secretion of cytokines by Th1 and Th2 cells. As shown in Figure 5F,G, SIgA and IgG levels were significantly reduced in the Cy group, and after the intervention of PS, the levels of immunoglobulins in mice were higher than those in the Cy group. In order to provide additional evidence of the immune-enhancing effects of PS in the spleen, a quantitative real-time polymerase chain reaction (qRT-PCR) was conducted to assess the transcriptional changes of *IL-10*, *TNF-α*, and *IL-6*. As depicted in Figure 5H–J, the mRNA expression levels of *IL-10*, *TNF-α*, and *IL-6* in the spleens of the Cy group were found to be considerably lower compared to those in the control group. In comparison to the Cy group, the PS treatment groups exhibited considerably elevated expression of *IL-10* and *TNF-α*. Additionally, the gene expression of *IL-6* was dramatically enhanced in the LPS group, while no significant differences were observed in the MPS and HPS groups. Therefore, the administration of PS has the potential to modulate the inflammatory response by impacting the production of various cytokines.

### 3.9. PS Modulated Oxidative Stress in Cyclophosphamide-Induced Immunosuppressed Mice

In order to explore the anti-oxidative properties of PS on immunosuppressed mice, the superoxide dismutase (SOD), nitric oxide (NO), malondialdehyde (MDA), and alanine aminotransferase (ALT) in the liver of mice were determined. As shown in Figure 6A–D, the SOD and NO levels of mice in the Cy group were significantly lower than those in the control group. The levels of SOD and NO were significantly increased in the MPS and HPS groups and the positive group, while the effect was not statistically significant in the LPS group. The content of MDA in the Cy group was significantly higher than that in the control group, while the contents of MDA in the MPS group and HPS group were significantly lower than that in the Cy group. The level of alanine aminotransferase (ALT) in the experimental group (Cy group) was found to be significantly higher compared to the control group, while PS at different doses resulted in a considerable reduction in ALT levels. These results indicate that PS improved oxidative stress and damage in the liver caused by cyclophosphamide.

### 3.10. PS Enhanced the Intestinal Barrier in Cyclophosphamide-Induced Immunosuppressed Mice

In order to observe the effect of PS on small intestinal mucosal injury, the small intestinal structure of mice was examined with a microscope, as shown in Figure 7. The villi in the control group were arranged in a neat and orderly manner, while they were destroyed in the Cy group. The intestinal structures in the PS treatment groups were partially restored. Compared with the control group, the Cy group had severe intestinal mucosal damage, with a decrease in the ratio of villous height to crypt depth, which was increased after PS treatment. The results of AB/PAS staining showed that cyclophosphamide could destroy the structural integrity of the intestinal tract of mice and reduce the goblet cells and mucus layer. PS reversed these changes and this indicates that PS had a protective effect on the intestinal injury caused by cyclophosphamide. The effect of PS on the IL-6/MyD88/TLR4 inflammatory pathway induced by cyclophosphamide was investigated by Western blotting. The expression of MyD88, IL-6, and TLR4 in the model group decreased (*p* < 0.001), indicating that cyclophosphamide leads to immunosuppression by inhibiting the expression of MyD88, IL-6, and TLR4. PS upregulated the protein expression of MyD88, TLR4, and IL-6 in the small intestine induced by Cy. Intestinal tight junction proteins play an important role in maintaining the stability of the intestinal environment, and the results showed that cyclophosphamide led to decreased expression of *zo-1*, *occludin*, and *mucin-2* proteins in the small intestine, while those proteins were improved by PS treatment, indicating that PS had a protective effect on the intestinal barrier of mice.

## 4. Discussion

Cyclophosphamide is metabolized in the liver to its active form to exert its anti-cancer function [37]; meanwhile, it could also induce various side effects including bone marrow suppression and immunosuppression [38]. The human immune system consists of various components, including immunological organs, immune cells, and immune molecules, which perform three major functions of surveillance, regulation, and defense. The spleen and thymus are the most important immune organs. *Polygonatum sibiricum* was recorded for medical use in ancient books more than 1000 years ago and is currently listed in the medicine and food homologous substances catalog. Saponin is one of the main active substances of *Polygonatum sibiricum* [39]. In this study, *Polygonatum sibiricum* saponin was examined for its immunomodulatory activity.

The spleen and thymus are pivotal organs associated with cellular and humoral immunity [40]. The observed augmentation in the mass of the spleen and thymus is indicative of the increased proliferation of lymphocytes inside these organs, and the atrophy of the immune organ may lead to a decrease in the body’s immune function [41]. Cyclophosphamide may cause atrophy, weight loss, and hematopoietic damage in mice, further leading to immunosuppression. The administration of immune boosters or nutritional supplements has the potential to augment the mass of these immune organs. Therefore, the spleen and thymus indices are commonly utilized as preliminary indicators of immune health in the search for immune-enhancing nutritional substances. In this study, the spleen, thymus, and immune indicators of mice were measured. The findings demonstrated that cyclophosphamide led to immunological organ atrophy and weight loss in mice. The blood WBC, RBC, HGB, HCT, PLT, LYM, and GRAN were significantly lower than those in the normal group and PS significantly alleviated these parameters, indicating that PS has an immunoprotective effect on cyclophosphamide-induced immunosuppressed mice by improving cyclophosphamide-induced hematopoietic impairment. Lymphocytes are important cellular components of the body’s immune response. T and B lymphocyte response to mitogens is a common immunological measurement used to assess lymphocyte response. We found that cyclophosphamide inhibited ConA-induced T lymphocyte proliferation and lipopolysaccharides-induced B lymphocyte proliferation [42,43]. Flow cytometry [44] results showed that the proportion of CD3^+^ T lymphocytes and CD355^+^ T cells in the Cy group was lower than that in the control group. These data suggest that PS could reverse cyclophosphamide-induced lymphocyte damage.

Macrophages play an important role in antitumor, antiviral, and autoimmune regulation [35]. Macrophages coordinate immune responses by secreting a variety of cytokines in response to stimuli. Phagocytosis and chemotaxis of macrophages could be modulated by altering transcriptional patterns so that invading intracellular parasitic bacteria could be eliminated [45,46]. It has been found that lncRNA GAS5 siRNA transfection promotes the uptake of E. coli by macrophages and regulates the phagocytosis of macrophages [47]. Our results showed that PS enhanced the phagocytic activity of macrophages against neutral red. Macrophage enzyme activity can reflect the activity status of macrophages and acid phosphatase and lactate dehydrogenase are indicators of macrophage activation [37,48]. Macrophages are involved in the entire immune process by clearing pathogens, presenting antigens, and releasing cytokines [49,50]. In this study, we showed that ACP and LDH in serum increased significantly after PS treatment, and PS increased the serum hemolysin content in mice and significantly reduced the rate of xylene-induced ear swelling, indicating that PS enhanced macrophage activity and humoral immunity in mice. Cells are stimulated by lipopolysaccharides, which promote the production of inflammatory cells and activate inflammatory pathways, leading to cell and tissue damage [51,52]. The balance between Th1 and Th2 cells is a crucial factor in the pathogenesis and progression of inflammatory disorders. Th1 cells are responsible for the secretion of interleukin-2 (IL-2) and interferon-gamma (IFN-γ), both of which contribute significantly to the activation and regulation of cell-mediated immune responses. Th2 cells secrete IL-4, IL-6, and IL-10, which mediate the immune response [53]. Interleukin-2 (IL-2) is synthesized by activated T cells and serves to stimulate their growth, proliferation, and differentiation. On the other hand, interferon-gamma (IFN-γ) facilitates the differentiation process and contributes to the development of cellular immune responses inside Th1 cells [54]. IL-2 and IFN-γ stimulate B cells to promote Ig production. PS, as a plant-derived saponin, increased the levels of TNF-α, IFN-γ, IL-4, IL-6, IgG, and SigA in the serum of cyclophosphamide-induced mice, and increased the mRNA expression of *IL-10*, *TNF-α*, and *IL-6*, which indicates that PS could restore the Th1/Th2 balance in immunosuppressed mice.

Cyclophosphamide can cause impaired immune function by causing oxidative stress. The dysregulation or excessive generation of reactive oxygen species (ROS) can result in tissue damage and impaired functionality across various tissues and organs [55]. Lipid peroxidation leads to reduced membrane fluidity, which leads to decreased immune function, increment of oxidative stress, and release of inflammatory factors [56]. SOD, MDA, and NO are intimately connected to the immunological system of the body. SOD can remove superoxide anions that are harmful to the body and maintain the body’s metabolic balance [57]. Nitric oxide (NO) can regulate the production and action of cytokines, participate in the proliferation, differentiation, and activation of immune cells, and play an important role in cellular immunity and humoral immunity. Studies have shown that *Cyclocarya paliurus* polysaccharide increases serum nitric oxide levels [58]. Our results show that PS reversed the decline in the activities of SOD and NO as well as the elevation of MDA in the Cy-treated mice. Therefore, our findings indicated that PS effectively relieved the oxidative stress induced by Cy partly through modulating antioxidative enzymes.

The intestine is the main digestive organ of the human body, with numerous immune cells, and is the largest immune organ of the human body [59]. Intestinal permeability is associated with the general health status of the body [60]. Lymphoid tissues respond immediately after being exposed to bacteria, secreting immunoglobulins to defend against foreign invaders [61]. Villi length and crypt depth are directly related to intestinal integrity, with increased crypt depth leading to decreased nutrient absorption capacity and longer villi enhancing nutrient absorption. The mice lost weight following cyclophosphamide treatment, which may suggest a decrease in the ability of the mice to absorb nutrients. The intestinal mucosal barrier is composed of intestinal epithelial cells and tight junction proteins, namely occludin, claudin, and zonula occludens (ZO-1, ZO-2, and ZO-3), as well as Mucin2. Transmembrane proteins, including claudin and occludin, facilitate the connection between two neighboring cells by means of extracellular loop interactions. The occurrence of abnormal claudin-1 expression is a prevalent characteristic observed in numerous disorders, indicating impairment of the intestinal barrier [62]. ZO-1 attaches claudin and occludin to the cytoskeleton, which is frequently utilized as a gauge for the health of the intestinal barrier [63]. Mucin-2 is produced by goblet cells and also plays a key role in maintaining intestinal mucosal barrier function [64,65]. SIgA is a distinctive secretory immunoglobulin in the gut, which plays an important role in maintaining intestinal homeostasis. The level of SIgA in the model group decreased after Cy treatment, which was recovered after PS treatment. Cyclophosphamide reduced the mRNA expression of *ZO-1*, *Occludin*, and *Mucin2* in the small intestine, while PS could significantly increase their expression, indicating a restorative effect of PS on intestinal mucosal injury.

## 5. Conclusions

In summary, PS could alleviate Cy-induced immunosuppression by maintaining the immune organ index, promoting lymphocyte proliferation and differentiation, modulating inflammatory cytokines, and relieving oxidative stress. Moreover, PS could strengthen compromised intestinal barriers caused by Cy by elevating the expression of junctional proteins. These data imply that *Polygonatum sibiricum* saponin could be potentially developed as an alternative strategy for mitigating chemotherapy-related immunosuppression.

## Figures and Tables

**Figure 1 foods-13-00934-f001:**
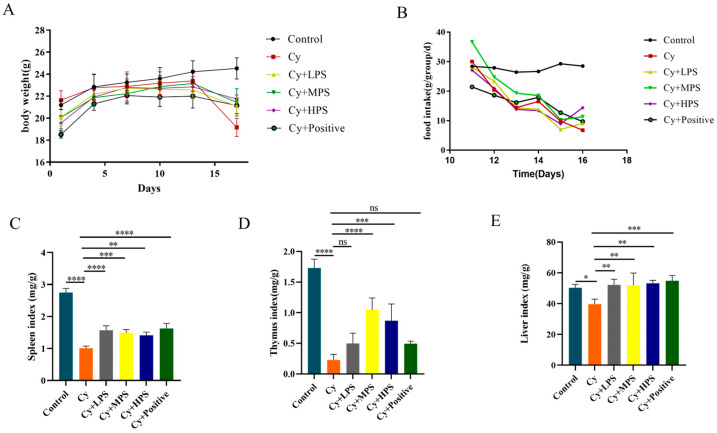
Effect of PS on the organ index. (**A**) Body weight. (**B**) Food intake (**C**) Spleen index (**D**) Thymus index (**E**) Liver index. Compared with the control. Data are shown as mean ± SEM (*n* = 12). * *p* < 0.05, ** *p* < 0.01, *** *p* < 0.001, **** *p* < 0.0001, ns means not statistically significant..

**Figure 2 foods-13-00934-f002:**
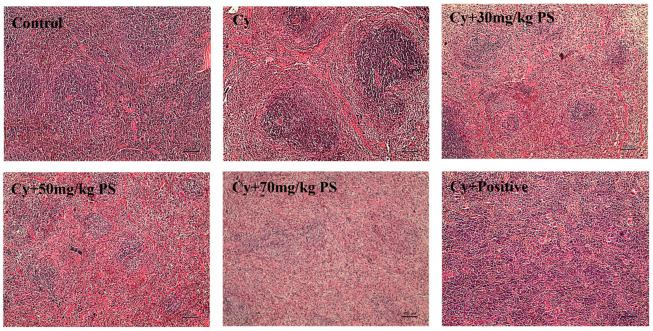
Histopathological changes in the spleen (scale bar, 100 µm).

**Figure 3 foods-13-00934-f003:**
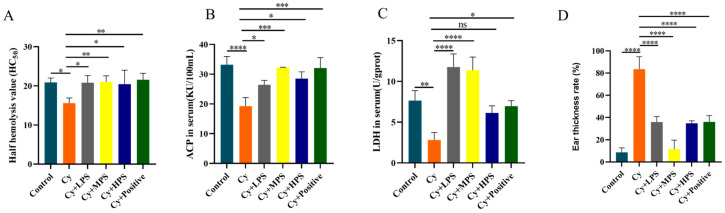
Effect of PS on hemolysin levels (*HC*50), acid phosphatase (ACP), lactate dehydrogenase (LDH) in mouse serum, and ear swelling rates in cyclophosphamide-induced mice. (**A**) Hemolysin values. (**B**) Acid phosphatase. (**C**) Lactate dehydrogenase (LDH). (**D**) Ear swelling rates (%). Compared with the control. Data are shown as mean ± SEM. * *p* < 0.05, ** *p* < 0.01, *** *p* < 0.001, **** *p* < 0.0001, ns means not statistically significant.

**Figure 4 foods-13-00934-f004:**
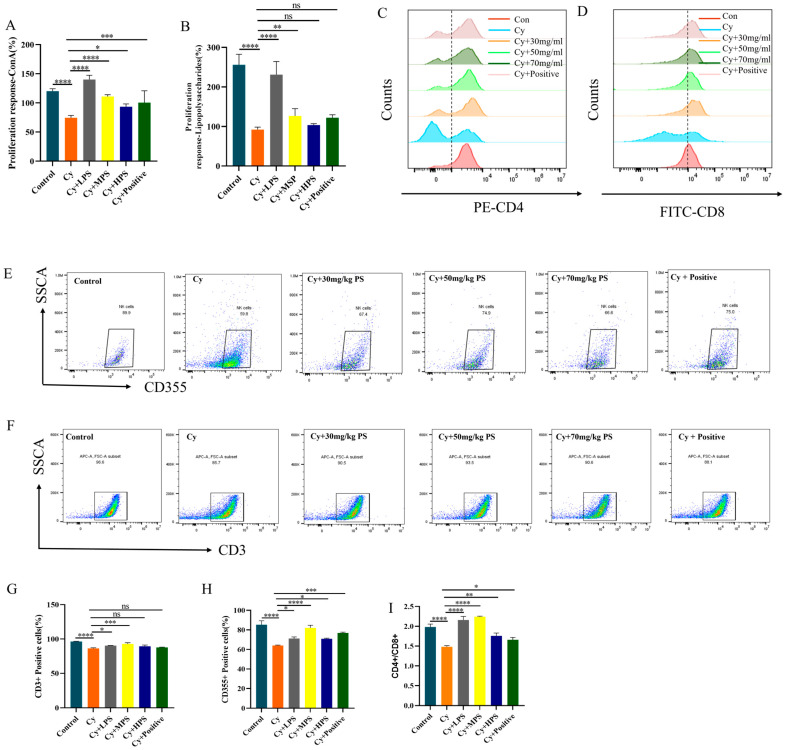
Effect of PS on cyclophosphamide-induced changes in spleen lymphocyte (*n* = 10). (**A**) ConA-induced T lymphocyte proliferation. (**B**) Lipopolysaccharides-induced B lymphocyte proliferation (*n* = 10). (**C**–**F**) CD3^+^, CD355^+^, CD4^+^, CD8^+^ flow cytometry results. Shades of color represent different cell densities. (**G**) CD3^+^ positive cell rates. (**H**) CD355^+^ positive cell rates. (**I**) CD4^+^/CD8^+^. Compared with the control. Data are shown as mean ± SEM. * *p* < 0.05, ** *p* < 0.01, *** *p* < 0.001, **** *p* < 0.0001, ns means not statistically significant.

**Figure 5 foods-13-00934-f005:**
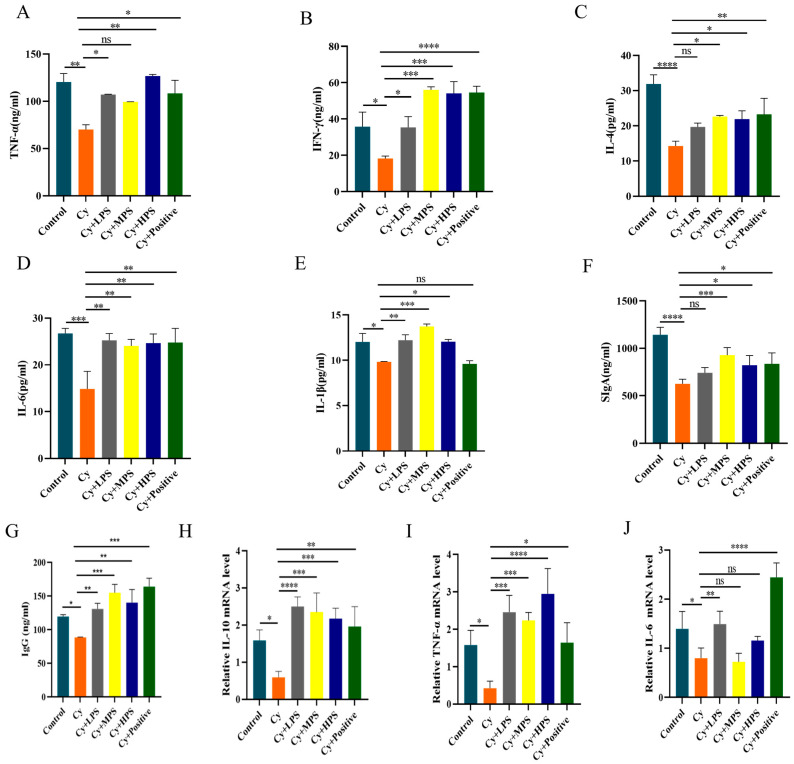
PS modulated cyclophosphamide-induced changes in serum cytokines and immunoglobulins and in spleen histology (**A**) TNF-α. (**B**) IFN-γ. (**C**) IL-4. (**D**) IL-6. (**E**) IL-1β. (**F**) SIgA. (**G**) IgG. (**H**) *IL-10* mRNA level. (**I**) *TNF-α* mRNA level. (**J**) *IL-6* mRNA level. Compared with the control. Data are shown as mean ± SEM. * *p* < 0.05, ** *p* < 0.01, *** *p* < 0.001, **** *p* < 0.0001, ns means not statistically significant.

**Figure 6 foods-13-00934-f006:**
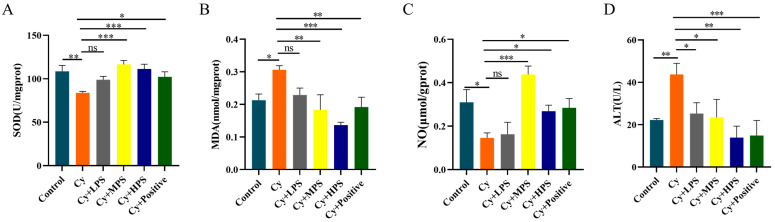
PS reversed the decrease of cyclophosphamide-induced antioxidant levels in mice (*n* = 5). (**A**) SOD. (**B**) MDA. (**C**) NO. (**D**) ALT. Compared with the control. Data are shown as mean ± SEM. * *p* < 0.05, ** *p* < 0.01, *** *p* < 0.001, ns means not statistically significant.

**Figure 7 foods-13-00934-f007:**
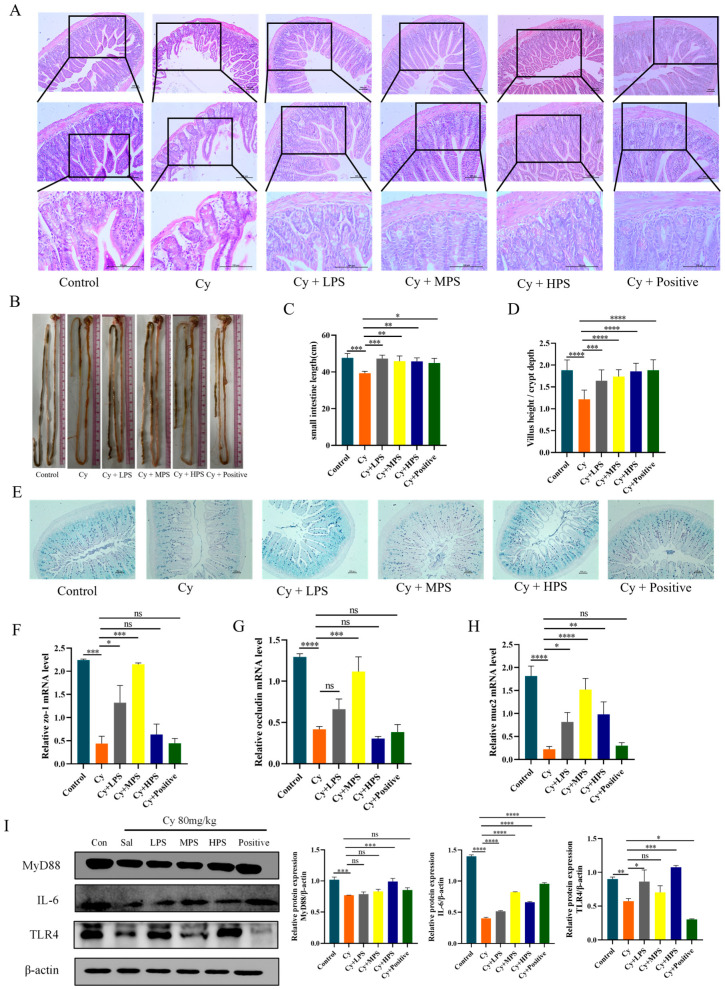
PS enhanced the intestinal barrier in Cyclophosphamide-induced immunosuppressed mice. (**A**) Images of hematoxylin and eosin-stained sections. (**B**) small intestine length. (**C**) The ratio of villi length to crypt depth. (**D**) small intestine length. (**E**) Images of AB/PAS-stained sections of the small intestine. Effect of MP on the relative expression of intestinal-related genes (**F**) *ZO-1*. (**G**) *Occludin*. (**H**) *Muc2*. (**I**) Effect of PS on MyD88/IL-6 pathway in the small intestine of immunosuppressed mice. Compared with the control. Data are shown as mean ± SEM. * *p* < 0.05, ** *p* < 0.01, *** *p* < 0.001, **** *p* < 0.0001, ns means not statistically significant.

**Table 1 foods-13-00934-t001:** Primer sequences (5′–3′) used for qRT-PCR.

Gene	Forward Sequence	Reverse Sequence
*IL-10*	GGTTGCCAAGCCTTATCGGA	GAGAAATCGATGACAGCGCC
*IL-6*	CTCAGCGCTGAGTTG	CCTGTAGCCCACGTCGTAGC
*TNF-α*	CGGGCAGGTCTACTTTGGAG	ACCCTGAGCCATAATCCCCT
*Mucin-2*	CCGGATCTATGCCGTTGCTA	TCCAGGTGGGTATCGAGTGT
*Zo-1*	ACCCGAAACTGATGCTGTGGATAG	AAATGGCCGGGCAGAACTTGTGTA
*Occludin*	TAGGGGCTCGGCAGGCTAT	CCGATCCATCTTTCTTCGGGT
*GAPDH*	TGTGTCCGTCGTGGATCTGA	TTGCTGTTGAAGTCGCAGGAG

**Table 2 foods-13-00934-t002:** Effect of PS on hematological parameters. WBC: white blood cell; RBC: red blood cell; HGB: hemoglobin; HCT: hematocrit; PLT: platelet; Lym: lymphocyte; Gran: granulocyte. Compared with the control. Data are shown as mean ± SEM (*n* = 5). * *p* < 0.05, ** *p* < 0.01, *** *p* < 0.001, **** *p* < 0.0001, ns means not statistically significant.

Hematological Parameter	Control	Cy	Cy + LPS	Cy + MPS	Cy + HPS	Cy + Positive
WBC (×10^9^/L)	9.55 ± 0.04 ****	8.75 ± 0.08	9.23 ± 0.13 ***	9.02 ± 0.07 *	9.17 ± 0.08 ***	9.11 ± 0.24 **
RBC (×10^12^/L)	13.05 ± 0.02 *	12.90 ± 0.03	13.07 ± 0.13 *	13.06 ± 0.04 *	13.00 ± 0.01 ^ns^	13.01 ± 0.06 ^ns^
HGB (g/dL)	3.24 ± 0.01 ****	3.08 ± 0.04	3.20 ± 0.01 ***	3.22 ± 0.03 ***	3.18 ± 0.01 **	3.15 ± 0.04 *
HCT (%)	54.47 ± 2.89 ***	38.65 ± 3.18	51.20 ± 1.22 *	57.30 ± 5.33 ****	49.33 ± 1.74 *	48.00 ± 3.62 *
PLT (×10^9^/L)	12.04 ± 0.13 ****	10.77 ± 0.03	11.35 ± 0.19 **	11.62 ± 0.14 ****	11.56 ± 0.06 ***	11.94 ± 0.20 ****
Lym (×10^9^/L)	9.43 ± 0.06 ****	8.77 ± 0.07	9.04 ± 0.13 **	8.97 ± 0.06 *	9.04 ± 0.05 **	9.19 ± 0.07 ****
Gran (×10^9^/L)	8.90 ± 0.09 **	8.45 ± 0.21	8.78 ± 0.15 ^ns^	8.48 ± 0 ^ns^	8.58 ± 0.16 ^ns^	8.67 ± 0.09 ^ns^

## Data Availability

The data presented in this study are available on request from the corresponding author.

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
