# Peer review of "Polygonatum sibiricum Saponin Prevents Immune Dysfunction and Strengthens Intestinal Mucosal Barrier Function in Cyclophosphamide-Induced Immunosuppressed BALB/c Mice"

_foods, 2024, doi:10.3390/foods13060934_

Round 1

Reviewer 1 Report

Comments and Suggestions for Authors

The article “ Polygonatum sibiricum Saponin Enhances Immunity and Intestinal Mucosal Barrier Function In Cyclophosphamide-Induced Immunosuppressed BALB/C Mice” focuses on studying the effect of Polygonatum sibiricum saponin (PS) on immune regulation. I am not sure that paper is in the scope of the Foods journal. 

This paper may be suitable for publication following minor revision. Authors should answer and resolve the following queries.

1. The first reference in the reference list is closer to the number than others. 

2. In the header of the paper is written "Foods 2021"

3. It is written, "Powder of Polygonatum sibiricum (50g) were extracted with 25 times the volume in 80% ethanol."

-Perhaps it would be better to write an exact volume of the solvent. 

4. Why do authors decide on in vivo studies? Are there results of any in vitro studies?

5. In 2.5. Xylene-induced inflammatory response is written

"To examine the acute inflammatory reactions caused by xylene in mice, 20ul amount.."

uL should be replaced with µL

6. 

In the last line, first paragraph page 7  is written :

"ACP activity and LDH activity were also increased in the HPS group, but there was no significant significance."

I suppose it should be written 

"ACP activity and LDH activity were also increased in the HPS group, but there was no significant difference."

7. 

Maybe change Figure 2B t and 2D. 

8.  In table  12 are near some values ##, I supouse it should be changed with *?

9. In 3.7. PS modulated inflammatory factors and immunoglobulin in Cyclophosphamide-induced immunosuppressed mice in line 7 is written "..group than in the MPS and HPS groups. but there was a.." The point should be replaced with a comma. 

10. Page 11, line 6 in  3.10. is written “the ratio of villous villus height” Please, check. 

11. “in vivo” should be written in italic

Author Response

Article title:Polygonatum sibiricum saponin prevents immune dysfunction and strengthens intestinal mucosal barrier function in Cyclophosphamide-induced immunosuppressed BALB/c mice

Editorial manuscript number:Foods-2853237

Dear Reviewers and Editors,

We are very grateful to the reviewers and editors for critical reading of the manuscript and many valuable suggestions. We have revised the manuscript according to the comments and marked changes in red in revised manuscript. The point-by -point response to the reviewers are as follows:

Response to Reviewer #1: The article “Polygonatum sibiricum Saponin Enhances Immunity and Intestinal Mucosal Barrier Function in Cyclophosphamide-Induced Immunosuppressed BALB/C Mice” focuses on studying the effect of Polygonatum sibiricum saponin (PS) on immune regulation. I am not sure that paper is in the scope of the Foods journal. This paper may be suitable for publication following minor revision. Authors should answer and resolve the following queries.

  1. The first reference in the reference list is closer to the number than others. 

---- Thanks. We've corrected the problem in the revised manuscript.

  1. In the header of the paper is written "Foods 2021"

---- We corrected this in the revised manuscript

  1. It is written, "Powder of Polygonatum sibiricum (50g) were extracted with 25 times the volume in 80% ethanol." Perhaps it would be better to write an exact volume of the solvent. 

---- Thanks for pointing this out. We have provided the exact volume in the revised manuscript. Powder of Polygonatum sibiricum (50g) were extracted with 1, 250 mL of 80% ethanol.

  1. Why do authors decide on in vivo studies? Are there results of any in vitro studies?

--- Thanks for the question. First, cyclophosphamide is metabolized in liver to its pharmalogically active form (4-hydroxycyclophosphamide (4OHCP) and other derivatives, which may together exert their effects on the body. Therefore, it is  difficult to choose which metabolite(s) for in vitro study; Second, since the immune system is composed of various components (such as cellular and humoral immunity), it is hard to examine holistically immunomodulatroy effect using one type of in vitro cell model. Therefore we followed other similar studies and focused on in vivo studies which are commonly utilized to examine immuno-enhancing effect of bioactive substances[1-5].

Just for your reference, we performed some related in vitro experiments to test how Polygonatum sibiricum saponin (PS) could impact lipopolysaccharide induced cell damage to RAW 264.7 macrophage cells. We found that PS exhibits no cytotoxicity at the concentration of 10-60 μg/mL, and could relieve mitochondria membrane potential loss induced by lipopolysaccharide, which is evidenced by the increased red fluorescence in PS treatment groups.

Figure PS alleviated lipopolysaccharide-induced damage in RAW264.7 macrophage cells. (A) Cell viability. (B) Mitochondrial membrane potential as indicated by fluorescence after JC-1 staining.

  1. In 2.5. Xylene-induced inflammatory response is written

"To examine the acute inflammatory reactions caused by xylene in mice, 20ul amount." uL should be replaced with µL

--- Thanks, we corrected this in the revised manuscript

  1. In the last line, first paragraph page 7 is written:

"ACP activity and LDH activity were also increased in the HPS group, but there was no significant significance." I suppose it should be written "ACP activity and LDH activity were also increased in the HPS group, but there was no significant difference."

--- Thanks, we had corrected this in the revised manuscript as suggested.

  1. Maybe change Figure 2B t and 2D. 

--- Thanks, we have changed the order as suggested (now Figure 3B and 3D in the revised manuscript).

  1. In table 12 are near some values ##, I suppose it should be changed with *?

--- Thanks, we corrected this in the Table 2 in the revised manuscript

  1. In 3.7. PS modulated inflammatory factors and immunoglobulin in Cyclophosphamide-induced immunosuppressed mice in line 7 is written ". group than in the MPS and HPS groups. but there was a.." The point should be replaced with a comma. 

--- Thanks, we corrected this as suggested.

  1. Page 11, line 6 in 3.10. is written “the ratio of villous villus height” Please, check. 

--- Thanks, we corrected this in the revised manuscript

  1. “in vivo” should be written in italic

--- Thanks, we corrected this in the revised manuscript

References

  1. Zheng S, Zheng H, Zhang R, Piao X, Hu J, Zhu Y, Wang Y: Immunomodulatory Effect of Ginsenoside Rb2 Against Cyclophosphamide-Induced Immunosuppression in Mice. Frontiers in pharmacology 2022, 13:927087.
  2. Liu JP, Wang J, Zhou SX, Huang DC, Qi GH, Chen GT: Ginger polysaccharides enhance intestinal immunity by modulating gut microbiota in cyclophosphamide-induced immunosuppressed mice. International journal of biological macromolecules 2022, 223(Pt A):1308-1319.
  3. Xu D, Hu J, Zhong Y, Zhang Y, Liu W, Nie S, Xie MJ: Effects of Rosa roxburghii&edible fungus fermentation broth on immune response and gut microbiota in immunosuppressed mice. 2024, 13(1):154-165.
  4. Huang J, Huang J, Li Y, Wang Y, Wang F, Qiu X, Liu X, Li H: Sodium Alginate Modulates Immunity, Intestinal Mucosal Barrier Function, and Gut Microbiota in Cyclophosphamide-Induced Immunosuppressed BALB/c Mice. Journal of agricultural and food chemistry 2021, 69(25):7064-7073.
  5. Ali MS, Lee EB, Quah Y, Birhanu BT, Suk K, Lim SK, Park SC: Heat-killed Limosilactobacillus reuteri PSC102 Ameliorates Impaired Immunity in Cyclophosphamide-induced Immunosuppressed Mice. Frontiers in microbiology 2022, 13:820838.
  6. Yu Y, Mo S, Shen M, Chen Y, Yu Q, Li Z, Xie J: Sulfated modification enhances the immunomodulatory effect of Cyclocarya paliurus polysaccharide on cyclophosphamide-induced immunosuppressed mice through MyD88-dependent MAPK/NF-κB and PI3K-Akt signaling pathways. Food research international (Ottawa, Ont) 2021, 150(Pt A):110756.

Reviewer 2 Report

Comments and Suggestions for Authors

The results of the manuscript entitled “Polygonatum sibiricum Saponin (PS) Enhances Immunity and Intestinal Mucosal Barrier Function in Cyclophosphamide-Induced Immunosuppressed BALB/C Mice” and authored by Zhao et al suggest that PS promoted lymphocyte proliferation and differentiation; upregulated Th1/Th2 cytokines, including inflammatory factors and immunoglobulin secretion levels. It also a restored the hematopoietic damage and induced the mRNA expression of small intestine tight junction protein, which also has a protective effect on the intestinal mucosal barrier. Authors finally argued that PS may potentially be developed as alternative strategy for enhancing immunity. Before exploring PS, it would be beneficial to include a broader overview of biomolecules and their overall impact on enhancing human health. The following studies investigate pathways that natural products manifest their effects on different diseases and should be integrated: https://doi.org/10.1186/s41936-020-00177-9, PMID: 33782460, PMID: 34662244, PMID: 32308651, PMID: 35517830. Discussing relevant patents would be useful. What happens if both PS and levamisole are applied together? The combined effect should have been assessed in a separate animal groups/s.

Detailed comments

·         Proofreading is REQUIRED.

·         The title should mention somewhere that this is a preventive study.

·         Abbreviation list would be useful if added.

·         How exactly were the animals euthanized? A detailed protocol is needed.

·         Figure 4K should be placed and cited much earlier in the manuscript. It is how the model system was established.

·         Original gels should be provided.

·         The following studies should be considered to enrich the discussion and update the references list: PMID: 36133773, https://patents.google.com/patent/US10912741B2/en, https://patents.google.com/patent/US10568873B1/en, PMID: 37568716, PMID: 37627094.

Comments on the Quality of English Language

Proofreading needed

Author Response

Article title:Polygonatum sibiricum saponin prevents immune dysfunction and strengthens intestinal mucosal barrier function in Cyclophosphamide-induced immunosuppressed BALB/c mice

Editorial manuscript number:Foods-2853237

Dear Reviewers and Editors,

We are very grateful to the reviewers and editors for critical reading of the manuscript and many valuable suggestions. We have revised the manuscript according to the comments and marked changes in red in revised manuscript. The point-by -point response to the reviewers are as follows:

Response to Reviewer:

Before exploring PS, it would be beneficial to include a broader overview of biomolecules and their overall impact on enhancing human health. The following studies investigate pathways that natural products manifest their effects on different diseases and should be integrated: https://doi.org/10.1186/s41936-020-00177-9, PMID: 33782460, PMID: 34662244, PMID: 32308651, PMID: 35517830. Discussing relevant patents would be useful. What happens if both PS and levamisole are applied together? The combined effect should have been assessed in a separate animal groups/s.

---- Thanks. We have added a description on the impact of natural products on enhancing human health in the revised manuscript and cited the literature suggested. Since levamisole hydrochloride has been commonly used as a positive control in similar studies exmining the immune-enhancing effects of natural products[2, 3, 4], we also select it as a positive drug which could restore suppressed immunity induced by cyclophosphamide. Although most studies did not examine the combined effect of positive drug and tested substance, we agree with the reviewer that it is interesting to explore whether there is a synergistic effect if PS and levamisole are applied together, considering they may exert their immuno-stimulatory effects through different pathways. We included this point in the revised discussion as a limitation of this study and will bear this insightful suggestion in mind in designing our future following studies.

Detailed comments

1.Proofreading is REQUIRED.

--- Thanks, we have revised the language throughout the manuscript and had it proofread by a native English speaker to minimize the language issues.

2.The title should mention somewhere that this is a preventive study.

--- Thanks, we have revised the title as‘Polygonatum sibiricum Saponin Prevents Immune Dysfunction and Strengthens Intestinal Mucosal Barrier Function in Cyclophosphamide-Induced Immunosuppressed BALB/C Mice’

3.Abbreviation list would be useful if added.

--- Thanks, we have added a list of abbreviation in the end of the revised manuscript as suggested.

4.How exactly were the animals euthanized? A detailed protocol is needed.

--- Thanks, we have supplemented the detailed steps for euthanasia in mice. Mice were fasted overnight after 15 days and provided with sterile water. Mice were intraperitoneally injected with sodium pentobarbital (80mg/kg) for anesthesia. Blood samples were taken by retrobulbar puncture and then mice were euthanized by cervical dislocation. Liver, spleen, thymus, small intestine, small intestine contents were then collected for further analysis.

5.Figure 4K should be placed and cited much earlier in the manuscript. It is how the model system was established.

--- Thanks. As suggested, we've moved the original Figure 4K to new Figure 2 in the revised manuscript.

6.Original gels should be provided.

---- Thanks. We have provided the original gel images (shown as follows) as separate documents in the submission system.

The following studies should be considered to enrich the discussion and update the references list: PMID:36133773, https://patents.google.com/patent/US10912741B2/en, https://patents.google.com/patent/US10568873B1/en, PMID: 37568716, PMID: 37627094.

---- Thanks. We have cited the papers and patents in the revised manuscript as suggested to enrich the discussion.

Reference

  1. Zheng S, Zheng H, Zhang R, Piao X, Hu J, Zhu Y, Wang Y: Immunomodulatory Effect of Ginsenoside Rb2 Against Cyclophosphamide-Induced Immunosuppression in Mice. Frontiers in pharmacology 2022, 13:927087.
  2. Liu JP, Wang J, Zhou SX, Huang DC, Qi GH, Chen GT: Ginger polysaccharides enhance intestinal immunity by modulating gut microbiota in cyclophosphamide-induced immunosuppressed mice. International journal of biological macromolecules 2022, 223(Pt A):1308-1319.
  3. Xu D, Hu J, Zhong Y, Zhang Y, Liu W, Nie S, Xie MJ: Effects of Rosa roxburghii&edible fungus fermentation broth on immune response and gut microbiota in immunosuppressed mice. 2024, 13(1):154-165.
  4. Huang J, Huang J, Li Y, Wang Y, Wang F, Qiu X, Liu X, Li H: Sodium Alginate Modulates Immunity, Intestinal Mucosal Barrier Function, and Gut Microbiota in Cyclophosphamide-Induced Immunosuppressed BALB/c Mice. Journal of agricultural and food chemistry 2021, 69(25):7064-7073.
  5. Ali MS, Lee EB, Quah Y, Birhanu BT, Suk K, Lim SK, Park SC: Heat-killed Limosilactobacillus reuteri PSC102 Ameliorates Impaired Immunity in Cyclophosphamide-induced Immunosuppressed Mice. Frontiers in microbiology 2022, 13:820838.
  6. Yu Y, Mo S, Shen M, Chen Y, Yu Q, Li Z, Xie J: Sulfated modification enhances the immunomodulatory effect of Cyclocarya paliurus polysaccharide on cyclophosphamide-induced immunosuppressed mice through MyD88-dependent MAPK/NF-κB and PI3K-Akt signaling pathways. Food research international (Ottawa, Ont) 2021, 150(Pt A):110756.

Round 2

Reviewer 2 Report

Comments and Suggestions for Authors

The revised version is much improved. No further comments from this reviewer.